

# Shoot and root traits of summer maize hybrid varieties with higher grain yields and higher nitrogen use efficiency at low nitrogen application rates

Wennan Su[1,2], Muhammad Kamran[2], Jun Xie[2], Xiangping Meng[1,2], Qingfang Han[1,2], Tiening Liu[1,2] and Juan Han[1,2]

[1] Key Laboratory of Crop Physio-ecology and Tillage Science in North-western Loess Plateau, Ministry of Agriculture / College of Agronomy, Northwest A&F University, Yangling, Shaanxi, China
[2] Key Laboratory of Agricultural Soil and Water Engineering in Arid and Semi-arid Areas, Ministry of Education / Institute of Water Saving Agriculture in Arid Areas of China, Northwest A&F University, Yangling, China

## ABSTRACT

Breeding high-yielding and nitrogen-efficient maize (*Zea* mays L.) hybrid varieties is a strategy that could simultaneously solve the problems of resource shortages and environmental pollution. We conducted a 2-year field study using four nitrogen application rates (0, 150, 225, and 300 kg N hm$^{-2}$) and two maize hybrid varieties (ZD958 and QS101) to understand the plant traits related to high grain yields and high nitrogen use efficiency (NUE). We found that ZD958 had a higher grain yield and nitrogen accumulation in the shoots at harvest as well as a higher NUE at lower nitrogen application rates (0 and 150 kg hm$^{-2}$) than QS101. The grain yields and NUE were almost identical for the two hybrid varieties at nitrogen application rates of 225 and 300 kg N hm$^{-2}$. Compared with QS101, ZD958 had higher above-ground and below-ground biomass amounts, a deeper root distribution, longer root length, root active absorption area, greater grain filling rate, and higher photosynthetic NUE than QS101 at lower nitrogen application rates. Our results showed that ZD958 can maintain a higher grain yield at lower nitrogen rates in a similar manner to N-efficient maize hybrid varieties. The selection of hybrids such as ZD958 with a deeper root distribution and higher photosynthetic NUE can increase the grain yield and NUE under low nitrogen conditions.

## INTRODUCTION

*Tester & Langridge (2010)* predicted that the global demand for food will increase by 70% by 2050, and thus there is an urgent need to address the problem of resource shortages in order to meet future human needs (*Good, Shrawat & Muench, 2004*). Nitrogen fertilizer is the largest input resource for agricultural production and it greatly improves crop yields (*Guo et al., 2010*). At present, the nitrogen fertilizer application rate far exceeds the appropriate range in China (*Fang et al., 2010*). The high nitrogen fertilizer application

Corresponding author
Qingfang Han,
hanqf88@nwafu.edu.cn

rate and low nitrogen use efficiency (NUE) have caused groundwater and air pollution in many regions of China (*Cui et al., 2010*; *Zhang et al., 2016*). Therefore, environmental and resource constraints mean that further increases in agricultural production should be achieved by increasing the NUE rather than higher nitrogen inputs (*Hawkesford, 2014*).

In 2005, China accounted for 38% of the global nitrogen consumptions but it only has 9% of the world's arable land (*FAOSTAT, 2011*). Nitrogen fertilization-induced increases in grain yields have been decreasing each year in China. Thus, the agricultural NUE decreased from 25 kg kg$^{-1}$ during 1958–1962 to 8 kg kg$^{-1}$ during 1997–2003, which was significantly below than the lower limit of the internationally recognized agricultural NUE range (10–30 kg kg$^{-1}$) (*Dobermann, 2005*). The NUE can be improved by adjusting the fertilization method employed, period, or types (*Abbasi, Tahir & Rahim, 2013*; *Zhao et al., 2013*). *Ciampitti & Vyn (2012)* and *Ciampitti & Vyn (2013)* concluded that modern varieties obtain relatively higher NUE levels than old varieties. However, the modern breeding process is usually conducted on fertile soil or soil with high nitrogen fertilizer inputs, and breeders have mainly focused on nitrogen absorption by plants and grain yields BerlinGallais2000, whereas they have neglected the performance of hybrid varieties under low nitrogen conditions.

The selection of N-efficient hybrid varieties (hybrids with higher grain yield under low N condition (*Wu et al., 2011a*; *Wu et al., 2011b*)) plays a significant role in improving the NUE. Differences in nitrogen uptake and utilization vary significantly among genotypes (*Kant, Bi & Rothstein, 2011*; *Bingham et al., 2012*), which are reflected in the responses of different varieties to nitrogen application, as well as in the distribution and utilization of nitrogen in crops (*Gallais & Coque, 2005*). However, there is still no general conclusion regarding the characteristics of high grain yield and high NUE plants under low nitrogen conditions.

Roots play a supporting and fixing role in the soil, but they are also important organs that allow plants to capture water and nutrients from the soil (*Lynch, 2013*; *White et al., 2013*). Previous studies of roots have investigated the possibility of further improving crop productivity and the NUE. The growth and distribution of roots are determined by genetic characteristics, and they are also influenced by environmental factors. The morphology of roots is closely related to the acquisition of soil resources and the development of plant shoots (*Mi et al., 2010*; *Lynch, 2013*; *Li et al., 2017*). The effects of various factors mean that roots respond to nitrogen in significantly different manners in field trials compared with culture experiments (*Wang et al., 2003*; *Tian et al., 2005*). The effects of nitrogen application on the roots are inconsistent under field conditions, where some studies suggest that nitrogen application can promote root growth, whereas others indicate that it can inhibit the growth of the root system (*Chen et al., 2015*; *Feng et al., 2016*). There are reports showing that several plant shoot traits could also be associated with grain yield and NUE in selecting for improving grain yield under low nitrogen conditions (*Banziger & Lafitte, 1997*; *Haegele et al., 2013*; *Chen et al., 2016*; *Talabi, Badu-Apraku & Fakorede, 2017*), such as leaf longevity (stay-green), anthesis-silking interval, leaf chlorophyll concentration, grain N accumulation and also number of kernels per ear. *Banziger et al. (2006)* and *Morosini et al. (2017)* suggested that the development of low-N tolerant and N-efficient genotypes can

result in significant breeding progress. However, there have been few systematic studies of the shoots and roots on the NUE in maize under field conditions.

In this study, we determined the characteristics of the shoots and roots in maize with high yield and high NUE under lower nitrogen application rates. These results obtained in this study provide insights into the mechanisms responsible for high grain yields and high NUE in maize, thereby providing a theoretical basis to allow breeders to improve the NUE without reducing the potential grain yield.

## MATERIALS AND METHODS

### Field experiments

Field experiments were conducted at the Agricultural Experimental Station of Northwest Agricultural and Forestry University (34°21′N, 108°10′E, 454.8 m altitude), Shaanxi province, northwest China during the maize growing season (June to October) of 2014 and 2015. The soil at the experimental site was loam soil with the following chemical properties in the top 60 cm soil layer at the start of the experiment: organic matter content, 14.3 g $kg^{-1}$; total N, 1.09 g $kg^{-1}$; Olsen-P, 9.4 mg $kg^{-1}$; and $NH_4OAc$-K, 127 mg $kg^{-1}$.

The experimental design was a randomized block with three replications. Each plot comprised one row with a length of 8 m and a row spacing of 60 cm, and the plants were spaced 25 cm apart. Two high-yielding maize varieties currently used for local production were selected as the typical contemporary maize hybrids: Zhengdan958 (ZD958: N-efficient hybrid) and Qiangsheng101 (QS101: N-inefficient hybrid). And the two maize hybrids have the same growth period. Plots were planted manually with two seeds per hill on June 17, 2014, and June 15, 2015. They were thinned to obtain the desired plant population (67500 pl. ha) at V3. The date of harvest was October 14, 2014 and October 15, 2015. Fertilizer N was sourced from urea (46% N), evenly split in the fractions of 1/2 at pre-sowing and side-banded deep (5 cm) into the soil on the sowing rows of 1/2 at twelve-leaf stage. In addition, 150 kg phosphorus ($P_2O_5$) $hm^{-2}$ as calcium superphosphate ($P_2O_5$ 16%) and 150 kg potassium ($K_2O$) $hm^{-2}$ as potassium sulfate ($K_2O$ 45%) were applied at pre-sowing. Irrigation was applied when the remaining 50% of urea (N 46%) was applied. Plots were kept free of weeds, insects, and diseases according to standard practices. Weather data were obtained from a local weather station approximately 100 m from the experimental field. The daily mean precipitation and temperature data during the two growing seasons are presented in Fig. 1. During the two growing seasons, the effective accumulated temperatures were 1617 °C and 1565 °C in 2014 and 2015, respectively, and the total precipitation levels were 217 mm and 299 mm.

### Sampling and measurements
#### Net photosynthetic rate (Pn) and photosynthetic nitrogen use efficiency (PNUE)

At anthesis from 10:00 to 12:00, three representative plants were selected from each plot and the Pn values of the ear leaves were measured with a portable infrared gas exchange-based photosynthesis analyzer system LI-6400 (LI-COR, Lincoln, NE, USA) while avoiding the midrib (prevent leaf chamber leakage), which was coupled with a standard red/blue LED
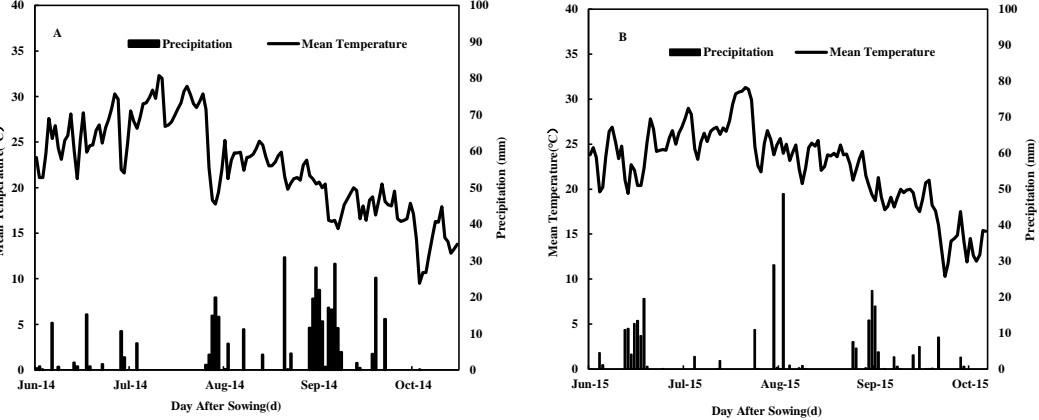

**Figure 1** Daily mean temperature, precipitation during the maize growing seasons in 2014 (A) and 2015 (B) at Yangling District, Shaanxi Province, China.

broadleaf cuvette (6400-02B, LI-COR), under a consistent controlled light intensity of 1,300 mmol m$^{-2}$ s$^{-1}$. After obtaining the photosynthesis measurements, the leaves were harvested to determine the leaf area. The veins were then removed and the remaining parts were mixed and heated at 105 °C for 30 min, before drying to constant weight at 70 °C. After determining the dry weight, the sample was ground into powder and the nitrogen concentration was obtained using the Kjeldahl method. Based on the leaf dry weight, leaf area, and nitrogen concentration, we calculated the leaf specific nitrogen (SLN (g N m$^{-2}$) = leaf nitrogen content (g) /leaf area (m$^2$)) and PNUE ((μmol CO$_2$ g$^{-1}$ N s$^{-1}$) = Pn of ear leaves (μmol m$^{-2}$ s $^{-1}$) /SLN (g N m$^{-2}$)) (*Sinclair & Horie, 1989*).

### Root length, dry weight and active absorption area

At anthesis, three representative and adjacent maize plants were selected in each plot the shoots were collected and the root system was excavated to a depth of 60 cm, where it was divided into three layers at every 20 cm using the soil profile method. In order to minimize any sampling and measurement errors, each root system was excavated from an area of 0.15 m$^2$(length = 0.6 m, width = 0.25 m). The excavated roots were washed and enclosed in a plastic bag, before scanning using a root scanner (with WinRHIZO scanning software). Then, the dry weight was determined after drying in an oven. Each scanned root image was processed using a root analysis program (Regent Instruments Inc. WinRHIZO Pro 2007d) to obtain the root length in each layer (*Chen et al., 2015*). The shoots were heated at 105 °C for 30 min and then dried to constant weight at 70 °C before weighing. Root active absorption area were determined by methylene blue dyeing method (*Zhang, Tan & Huang, 1994*).

### Grain filling characteristics

During 2014 and 2015, beginning 4 days after anthesis, three tagged ears were sampled per plot every 4 days until the grain reached physiological maturity. The grain in each row was selected and stripped, before counting the total number of kernels removed from each ear.

The kernels were dried until constant weight in an oven at 75 °C and then weighed with a balance.

### Shoot biomass and grain yield

At maturity (when black layer formation was complete in all plants), the grain yield was determined using plants from four undisturbed rows, where the plants on each side of the plot were discarded to avoid border effects, and grain yield was expressed at 14% moisture. Twenty plants in each plot were used to determine the kernel number and 100-grain dry weight. The dry weights of shoots from each plant were determined by oven-drying to constant weight at 75 °C. The plant nitrogen concentration was determined with the micro-Kjeldahl method (as described by *Nelson & Sommers (1973)*), distillation, and titration to calculate the shoot N uptake. The NUE was calculated according to the methods described by *Xue et al. (2013)* as well as the following.

Agronomic efficiency of nitrogen fertilizer (AEN, kg kg$^{-1}$) = (grain yield with nitrogen application − grain yield without nitrogen application)/amount of nitrogen applied

Nitrogen partial factor productivity (PFPN, kg kg$^{-1}$) = grain yield /amount of nitrogen applied

Nitrogen utilization efficiency (NUtE, kg kg$^{-1}$) = grain yield /nitrogen uptake by shoots at maturity

### Statistical analysis

The effects of treatments (hybrid variety and nitrogen application rate) and their interactions were compared with analysis of variance (ANOVAs) for related traits performed in 2014 and 2015 using SPSS 18.0 software (SPSS Inc., Chicago, IL, USA). Multiple comparisons were performed using Duncan's multiple range test and differences were considered statistically significant at $P < 0.05$.

We analyzed the kernel dry weight dynamics using the logistic equation (*Gu et al., 2001*) in Eq. (1):

$$W = \frac{A}{\left(1 + Be^{-Ct}\right)^{\frac{1}{D}}},$$

Where $W$ is the measured kernel dry weight (mg), $t$ is the number of days after anthesis, $A$ is the final kernel weight, $B$ is the initial value parameter, and $C$ is the growth rate parameter. We calculated the following grain filling characteristic parameters (*Wang et al., 2014*).

Days required to reach the maximum grain filling rate (Tmax, d) (2)

$$Tmax = \frac{\ln B}{C}$$

Grain weight at the maximum grain filling rate (Wmax, mg) (3)

$$Wmax = \frac{A}{2}$$

Maximum grain-filling rate (Gmax, mg grain$^{-1}$ day$^{-1}$) (4)

$$Gmax = C \times Wmax - \frac{C \times Wmax^2}{A}$$

**Table 1** Analysis-of-variance of grain yield, yield components, shoot nitrogen uptake and nitrogen use efficiency of maize between/among hybrid variety and nitrogen application rates in 2014 and 2015.

| Year | Source of variation | Grain yield (t hm$^{-2}$) | 100-dry grain weight (g) | Grain number (ear$^{-1}$) | Shoot nitrogen uptake (kg hm$^{-2}$) | AEN | PFPN | NUtE |
|------|---------------------|--------------------------|--------------------------|---------------------------|--------------------------------------|-----|------|------|
| 2014 | Hybrid variety (H) | ** | ** | ns | ** | * | * | ** |
|      | N rates(N) | ** | ** | ** | ** | ** | ** | ** |
|      | H*N | * | * | ns | ‡ | * | * | * |
| 2015 | Hybrid variety(H) | * | * | ns | ** | * | * | ** |
|      | N rates (N) | ** | ** | ** | ** | ** | ** | ** |
|      | H*N | ‡ | ‡ | ns | * | * | * | * |

Notes.

ns, Not significant at the $P = 0.05$ level.
*Significant at the $P = 0.05$ level.
**Significant at the $P = 0.01$ level.
‡Significant at the $P = 0.1$ level.
AEN, agronomic efficiency of nitrogen fertilizer (kg kg$^{-1}$); PFPN, nitrogen partial factor productivity (kg kg$^{-1}$); NUtE, nitrogen utilization efficiency (kg kg$^{-1}$).

Final grain filling time (time taken for the grain weight to reach 99% (T0.99, days)) (5)

$$T0.99 = \frac{\ln B + 4.59512}{C}$$

Logistic equation for calculating the grain weight when t = T0.99 (W1, mg) (6)

$$W1 = \frac{A}{\left(1 + Be^{-CT0.99}\right)^{\frac{1}{D}}}$$

Average grain filling rate (Gave, mg grain$^{-1}$ day$^{-1}$) (7)

$$Gave = \frac{W1}{T0.99}$$

The curves of the kernel dry weight dynamics were performed using SPSS 18.0 (SPSS Inc., Chicago, IL, USA).

# RESULTS

## Variance analysis

Table 1 shows the grain yield, yield components, shoot nitrogen uptake, and NUE in both years. The hybrid varieties, nitrogen application rate, and their interactions had significant effects ($P < 0.05$) on the different variables (except for the kernel number). Measurements of other indicators with similar results are not presented.

## Grain yield and nitrogen use efficiency (NUE)

As shown in Table 2, at lower nitrogen application rates (N0 and N150), the grain yields were significantly higher for ZD958 than QS101 at the same N rate ($P < 0.05$) There were no significant differences ($P \geq 0.05$) in the grain yields from both hybrid varieties under normal nitrogen application rates and higher application rates (N225 and N300) in both years. The grain yield was higher at lower nitrogen rates with ZD958, which was mainly attributed to higher 100-grain weight rather than the number of kernels compared with QS101 (Table 2).

Similar to the grain yield, the shoot nitrogen uptake, NUtE, and PFPN were significantly higher for ZD958 than QS101 at lower nitrogen application rates. These indicators did not

**Table 2** Grain yield and yield components of maize hybrid varieties under four nitrogen application rates in 2014 and 2015.

| N rate | Hybrid variety | 2014 | | | 2015 | | |
|---|---|---|---|---|---|---|---|
| | | 100-dry grain weight (g) | Grain number (ear$^{-1}$) | Grain yield (t hm$^{-2}$) | 100-dry grain weight (g) | Grain number (ear$^{-1}$) | Grain yield (t hm$^{-2}$) |
| N0 | QS101 | 28.5c | 417.56c | 7.36c | 27.73d | 428.67d | 7.71c |
| N150 | QS101 | 28.65c | 488.81ab | 8.44b | 30.97b | 496.45b | 8.74b |
| N225 | QS101 | 31.34a | 506.22ab | 9.65a | 32.64a | 523.49ab | 10.34a |
| N300 | QS101 | 31.17a | 502.81ab | 9.76a | 32.28a | 495.3b | 10a |
| N0 | ZD958 | 30.02b | 456.95cb | 8.45b | 29.34c | 461.73c | 8.69b |
| N150 | ZD958 | 31.74a | 523.87a | 9.57a | 32.51a | 519.96ab | 9.78a |
| N225 | ZD958 | 31.44a | 508.64ab | 9.66a | 32.68a | 558.13a | 10.26a |
| N300 | ZD958 | 31.58a | 544.07ab | 9.72a | 32.87a | 518.93ab | 10.06a |

**Table 3** Shoot nitrogen uptake and nitrogen use efficiency of two maize hybrid varieties under four nitrogen application rates in 2014 and 2015.

| N rate | Hybrid variety | 2014 | | | | 2015 | | | |
|---|---|---|---|---|---|---|---|---|---|
| | | Shoot N uptake (kg hm$^{-2}$) | AEN | PFPN | NUtE | Shoot N uptake (kg hm$^{-2}$) | AEN | PFPN | NUtE |
| N0 | QS101 | 115f | – | – | 54.91b | 117f | – | – | 56.73b |
| N150 | QS101 | 150d | 7.18b | 56.27b | 48.47d | 148d | 6.86b | 58.25b | 50.66d |
| N225 | QS101 | 190b | 10.14a | 42.87c | 43.71e | 186b | 11.69a | 45.95c | 47.8e |
| N300 | QS101 | 206a | 7.99b | 32.54c | 40.89e | 189a | 7.65b | 33.34c | 45.42e |
| N0 | ZD958 | 121e | – | – | 59.78a | 126e | – | – | 59.19a |
| N150 | ZD958 | 156c | 7.46b | 63.8a | 52.65c | 157c | 7.26b | 65.17a | 53.65c |
| N225 | ZD958 | 191b | 5.35b | 42.91c | 43.39e | 184b | 7b | 45.6c | 47.94e |
| N300 | ZD958 | 205a | 4.22c | 32.4c | 40.84e | 191a | 4.57c | 33.52c | 45.19e |

**Notes.**
AEN, agronomic efficiency of nitrogen fertilizer (kg kg$^{-1}$); PFPN, nitrogen partial factor productivity (kg kg$^{-1}$); NUtE, nitrogen utilization efficiency (kg kg$^{-1}$).

differ significantly between the two hybrid varieties at N225 and N300 (Table 3). The AEN values were similar for the two hybrids under N150. ZD958 had lower AEN values than QS101 under the N225 and N300 treatments, mainly because QS101 had a much lower grain yield than that ZD958 with nil nitrogen application. Thus, at low nitrogen rates, ZD958 with high grain yield and NUE defined as an N-efficient maize hybrid, whereas QS101 with low grain yield and NUE was defined as an N-inefficient maize hybrid. This confirms the choice of both hybrids.

## Shoots and roots

Figure 2 shows the shoot and root biomasses and root/shoot ratios at anthesis. At lower nitrogen application rates, the shoot and root biomasses were significantly higher with ZD958 than QS101. These parameters did not differ significantly between the two hybrid varieties at N225 and N300 in both years.

At maturity, the shoot biomass was significantly greater with ZD958 than QS101 under the N0 and N150 treatments. Under N225 and N300, the shoot biomass did not differ significantly between the two hybrid varieties. At the same nitrogen rate, there was no

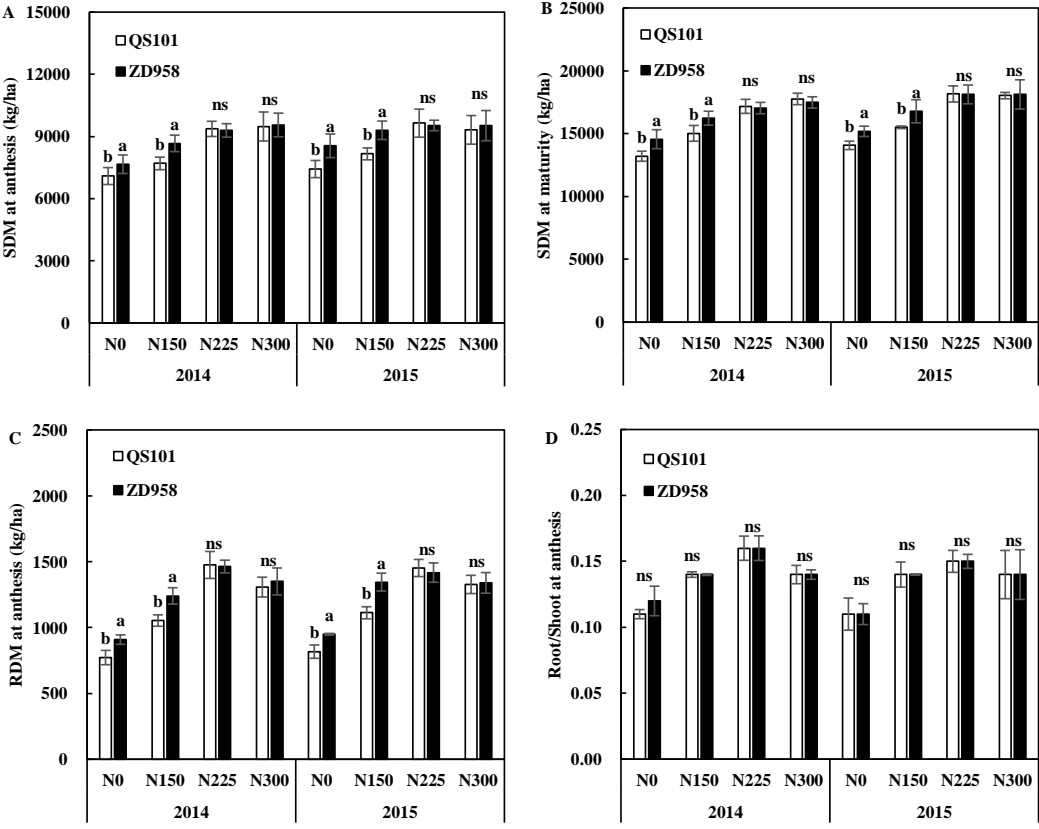

**Figure 2** **Shoot dry matter (SDM) at anthesis (A) and maturity (B), root dry matter (RDM, C) and root/shoot at anthesis (D) of maize under various nitrogen rates in 2014 and 2015.** Vertical bars represent ±standard error of the mean ($n = 3$) where these exceed the size of the symbol. Different lowercase letters above the column indicate statistical significance at the $P = 0.05$ level within the same N rate, and ns means not significant at the $P = 0.05$ level.

significant difference in the root/shoot ratio between the two varieties. The root/shoot ratio increased initially and then decreased as the nitrogen application rate increased for the same hybrid variety.

Figures 3–5 show the root dry weights, root lengths and root active absorption area for the two hybrid varieties under different nitrogen application rates in both years. Under the N0 and N150 treatments, the root dry weight and root length in the 0–60 cm soil depth were significantly higher for ZD958 than QS101. The root dry weight and root length in the 0–60 cm soil depth did not differ significantly between the two hybrid varieties under the N225 and N300 treatments, and they also did not differ significantly in the same hybrid between the N225 and N300 treatments. Moreover, the differences in the root dry weight and root length in the 0–20 cm soil depth were not significant between the two hybrid varieties under all of the nitrogen rates, whereas they differed significantly in the 20–60 cm soil depth. Thus, the differences between the two hybrid varieties under the N0 and N150 treatments were related to differences in the deeper roots. The root dry weight and root length increased as the nitrogen application rate increased, and then decreased.

Compared with the N225 treatment, the root dry weight and root length in the 0–60 cm soil depth were significantly different in the N300 treatment because the root dry weight in the 20–60 cm soil depth was significantly lower compared with that in the N225 treatment for both hybrid varieties (Figs. 3 and 4). Similar to root dry weight and root length, root active absorption area of QS101 in the 20–40 cm and 40–60 cm soil layers decreased were significantly lower than those of ZD958 under N0 and N150 treatments (Fig. 3–5).

### Leaf specific nitrogen (SLN), net photosynthetic rate (Pn), and photosynthetic nitrogen use efficiency (PNUE)

The SLN, Pn, and PNUE results are shown in Fig. 5. Under the N0 and N150 treatments, SLN was significantly higher for ZD958 than QS101, but there were no significant difference between the two hybrid varieties under N225 and N300. Similar to SLN, the Pn and PNUE values were significantly higher for ZD958 than QS101 under the N0 and N150 treatments. The differences between the two hybrid varieties were not significant for Pn and PNUE in N225 and N300. SLN and Pn tended to increase with the nitrogen application rate, whereas PNUE exhibited a decreasing trend (Fig. 6).

### Grain filling characteristics

Table 4 shows the grain filling parameters for the two hybrid varieties under different nitrogen application rates. Under N0 and N150 treatments, ZD958 exhibited a higher maximum grain filling rate (Gmax, mg grain$^{-1}$ d$^{-1}$), average grain-filling rate (Gave, mg grain$^{-1}$ d$^{-1}$), and kernel weight at the maximum grain filling rate (Wmax, mg) than QS101. QS101 required more days to reach the maximum grain filling rate (Tmax, d) than ZD958.

## DISCUSSION

Many studies have shown that the application of nitrogen fertilizer can improve the photosynthetic capacity, nitrogen content, total biomass, and grain yield but decrease the NUE (*Peng et al., 2016*; *Hammad et al., 2017*). Similar to previous studies, our results showed that the application of N300 increased the grain yield by 2–15% compared with N150. The PFPN value under the N300 treatment was decreased by 42–49% compared with the N150 treatment in both hybrid varieties (average of two years). Previous studies have rarely reported details of the plant traits related to high grain yields and high NUE levels in maize hybrid varieties under low nitrogen application rates.

It is well known that the NUE can be simply defined as the increase in the amount of nitrogen that plants obtain from soil, or it can be defined as the more efficient use of absorbed nitrogen (*Garnett, Conn & Kaiser, 2009*). Breeding hybrid varieties with both traits (especially under nitrogen-deficient conditions) will be beneficial for agricultural sustainability (*Garnett, Conn & Kaiser, 2009*; *Kant, Bi & Rothstein, 2011*). In this study, ZD958 obtained a higher yield than QS101 at low nitrogen rates (N0 and N150) and it absorbed more nitrogen from the soil while maintaining a higher NUE. These results indicate the possibility of developing maize hybrid varieties with both higher nitrogen uptake efficiencies and higher NUE levels. The grain yield increases of hybrid varieties with

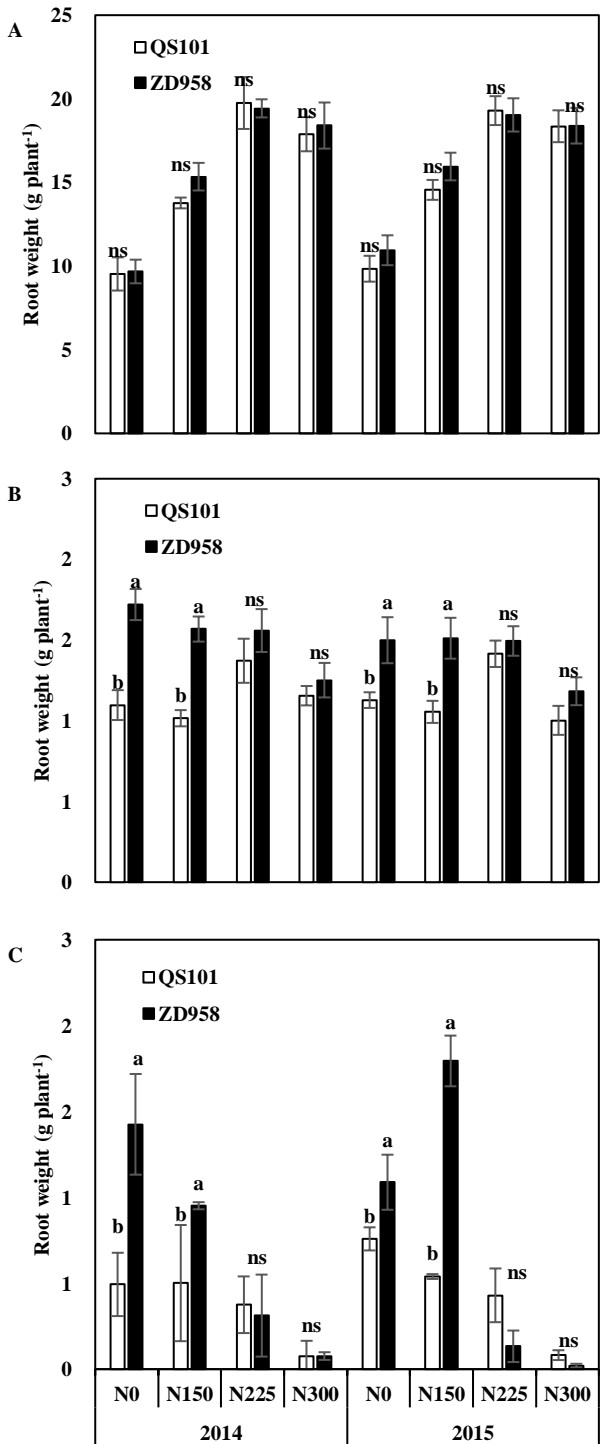

**Figure 3** **Root dry weight of maize in 0–20 cm (A), 20–40 cm (B) and 40–60 cm (C) soil layer at anthesis of two hybrid variety under various nitrogen (N) application rates in 2014 and 2015.** Data are averages observed for three replications. Vertical bars represent ±standard error of the mean ($n = 3$) where these exceed the size of the symbol. Different lowercase letters above the column indicate statistical significance at the $P = 0.05$ level within the same N rate, and ns means not significant at the $P = 0.05$ level.

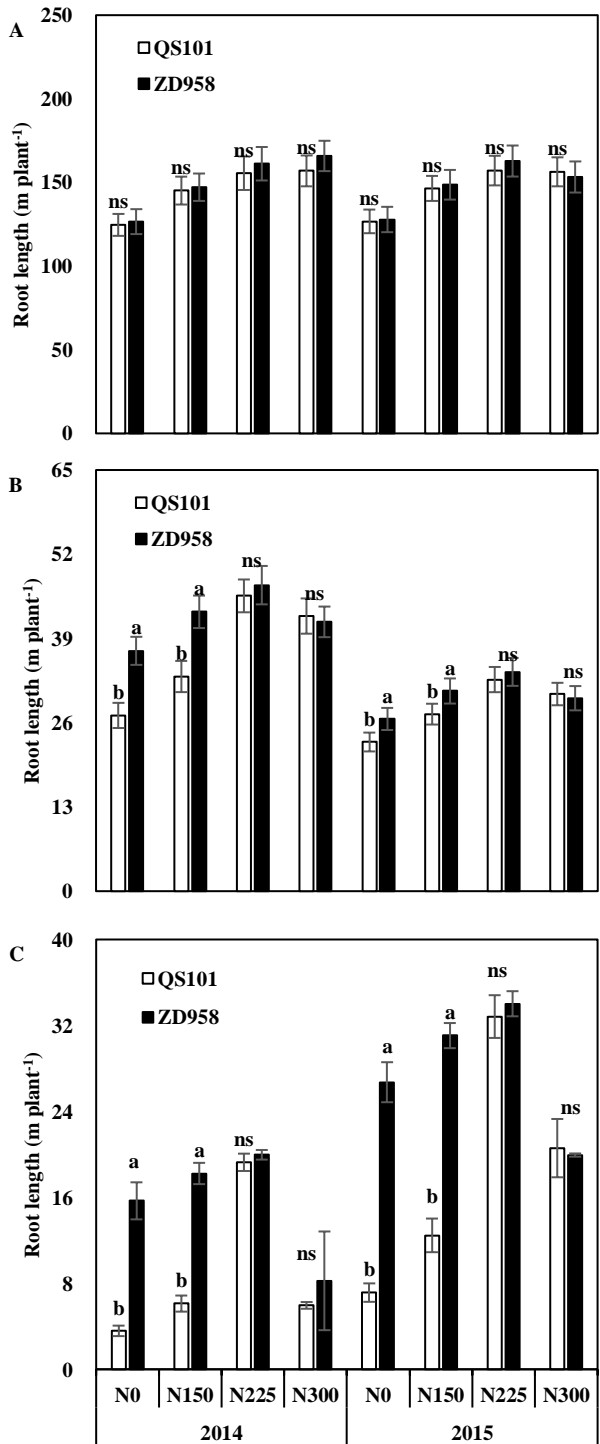

**Figure 4   Root length of maize in 0–20 cm (A), 20–40 cm (B) and 40–60 cm (C) soil layer at anthesis of two hybrid variety under various nitrogen (N) application rates in 2014 and 2015.** Data are averages observed for three replications. Vertical bars represent ±standard error of the mean ($n = 3$) where these exceed the size of the symbol. Different lowercase letters above the column indicate statistical significance at the $P = 0.05$ level within the same N rate, and ns means not significant at the $P = 0.05$ level.

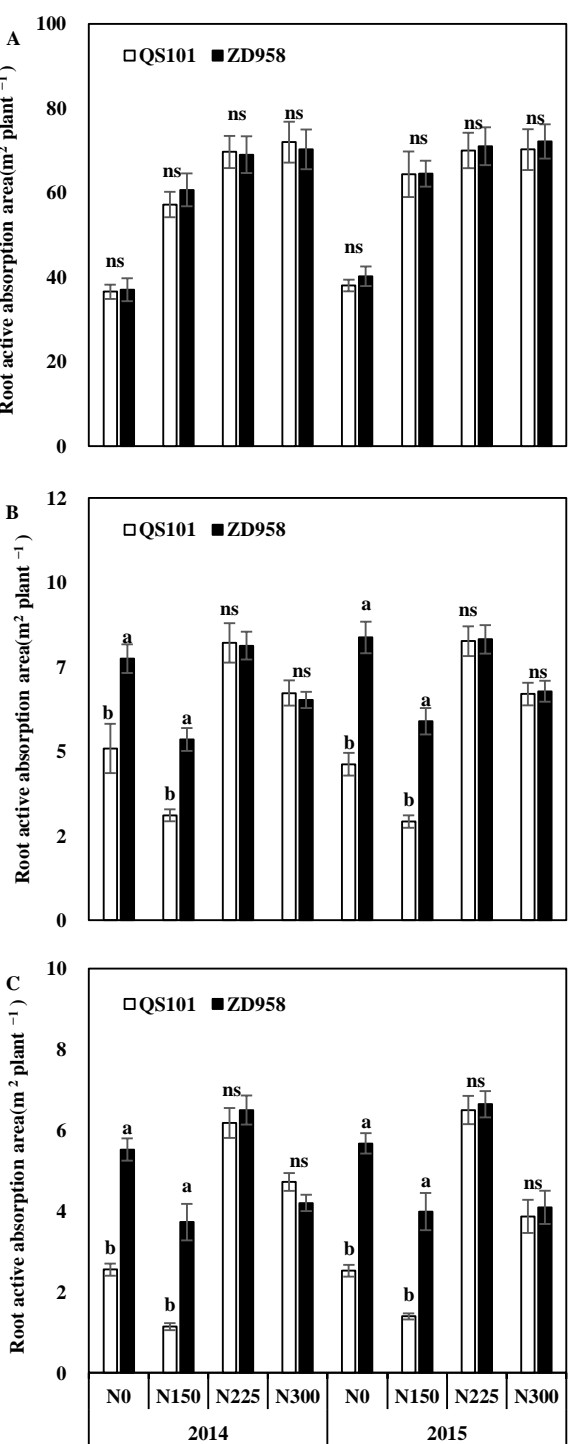

**Figure 5   Root active absorption area of maize in 0–20 cm (A), 20–40 cm (B) and 40–60 cm (C) soil layer at anthesis of two hybrid variety under various nitrogen (N) application rates in 2014 and 2015.** Data are averages observed for three replications. Vertical bars represent ±standard error of the mean ($n = 3$) where these exceed the size of the symbol. Different lowercase letters above the column indicate statistical significance at the $P = 0.05$ level within the same N rate, and ns means not significant at the $P = 0.05$ level.

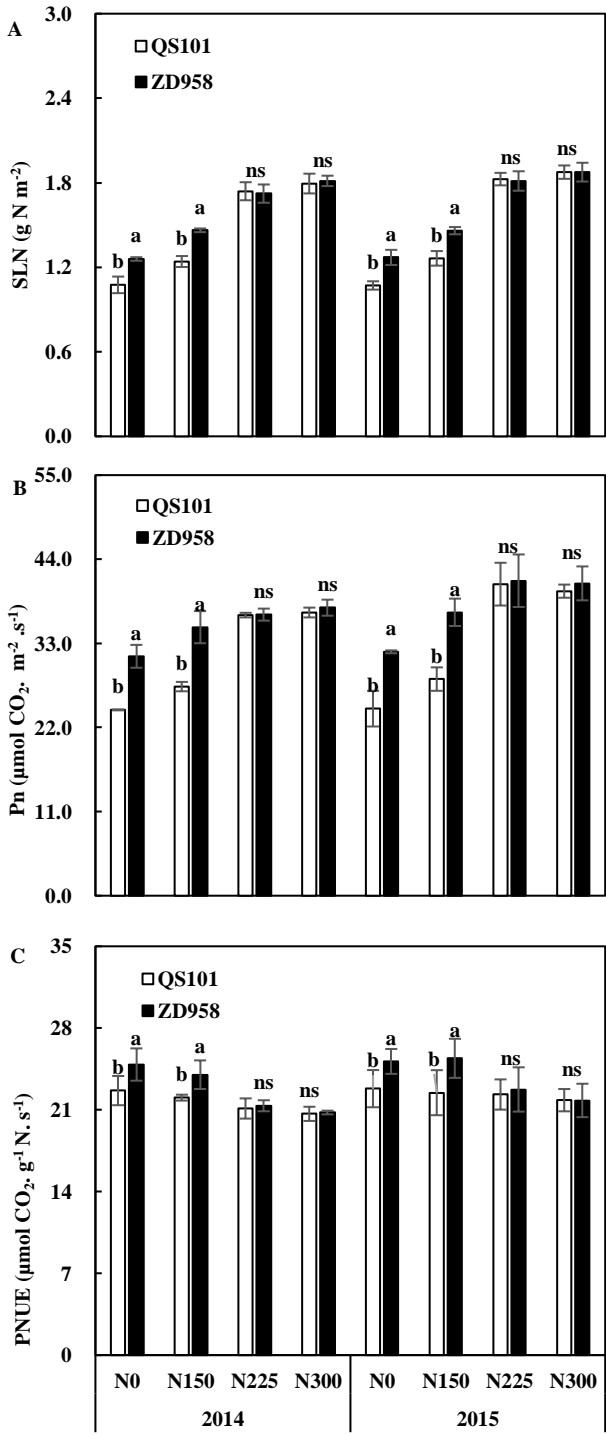

**Figure 6  Specific leaf nitrogen (SLN, A), net photosynthetic rate (Pn, B) and photosynthesitic nitrogen use efficiency (PNUE, C) of two maize hybrid varieties under four nitrogen rates at anthesis in 2014 and 2015.** Vertical bars represent ±standard error of the mean ($n = 3$) where these exceed the size of the symbol. Different lowercase letters above the column indicate statistical significance at the $P = 0.05$ level within the same N rate, and ns means not significant at the $P = 0.05$ level.

**Table 4  Shoot nitrogen uptake and nitrogen use efficiency of two maize hybrid varieties under four nitrogen application rates in 2014 and 2015.**

| Year | Parameters | N0-QS | N150-QS | N225-QS | N300-QS | N0-ZD | N150-ZD | N225-ZD | N300-ZD |
|------|-----------|-------|---------|---------|---------|-------|---------|---------|---------|
| 2014 | A    | 21.85  | 25.72 | 28.09  | 28.46  | 25.44 | 28.15 | 28.22 | 28.38 |
|      | B    | 108.90 | 97.59 | 104.93 | 123.51 | 59.33 | 61.62 | 60.40 | 68.62 |
|      | C    | 0.21   | 0.20  | 0.20   | 0.19   | 0.19  | 0.19  | 0.19  | 0.19  |
|      | Tmax | 22.59  | 22.95 | 23.51  | 24.95  | 21.07 | 21.32 | 21.85 | 22.43 |
|      | Gmax | 1.13   | 1.28  | 1.39   | 1.37   | 1.23  | 1.36  | 1.32  | 1.34  |
|      | Wmax | 10.92  | 12.86 | 14.05  | 14.23  | 12.72 | 14.07 | 14.11 | 14.19 |
|      | Gave | 0.48   | 0.55  | 0.60   | 0.58   | 0.56  | 0.62  | 0.60  | 0.60  |
|      | T0.99| 44.72  | 45.97 | 46.72  | 48.75  | 44.79 | 45.10 | 46.33 | 46.80 |
| 2015 | A    | 22.95  | 26.99 | 31.49  | 31.52  | 26.56 | 29.86 | 31.25 | 31.13 |
|      | B    | 64.47  | 62.79 | 68.15  | 79.76  | 42.08 | 44.72 | 44.32 | 50.17 |
|      | C    | 0.19   | 0.18  | 0.18   | 0.18   | 0.18  | 0.18  | 0.18  | 0.18  |
|      | Tmax | 22.36  | 22.78 | 23.41  | 24.94  | 20.79 | 21.08 | 21.65 | 22.25 |
|      | Gmax | 1.07   | 1.23  | 1.42   | 1.38   | 1.19  | 1.35  | 1.37  | 1.37  |
|      | Wmax | 11.48  | 13.49 | 15.75  | 15.76  | 13.28 | 14.93 | 15.62 | 15.57 |
|      | Gave | 0.48   | 0.56  | 0.64   | 0.61   | 0.57  | 0.63  | 0.65  | 0.64  |
|      | T0.99| 47.02  | 48.06 | 48.88  | 51.12  | 46.34 | 46.58 | 47.89 | 48.36 |

Notes. N0-QS, N150-QS, N225-QS, and N300-QS represent hybrid variety QS101 under 0,150,225, and 300 kg hm$^{-2}$ nitrogen application rates respectively; N0-ZD, N150-ZD, N225-ZD, N300-ZD represent hybrid variety ZD958 under 0,150,225, and 300 kg hm$^{-2}$ nitrogen application rates respectively.

Tmax, the days needed for reaching the maximum grain filling rate (d); Wmax, the grain weight at the maximum grain filling rate (mg); Gmax, the maximum grain-filling rate (mg grain$^{-1}$ d$^{-1}$); T0.99, the final grain filling time (the time of grain weight reaches 99%, d); Gave, the average grain-filling rate (mg grain$^{-1}$ d$^{-1}$).

high-yielding and high NUE range from 8% to 10%, and they can reduce the application of nitrogen fertilizer by 16%, thereby demonstrating that it is feasible to cultivate new high yield and high NUE maize varieties (*Chen et al., 2013*). *Adu et al. (2018)* also suggested maize grain yield could increase under low-N supply at the same time maintaining the grain yield potential under high-N conditions is achievable when NUE genotypes are adopted.

Most previous studies of the NUE in maize were conducted in potting conditions and few studies investigated the shoot and root systems in hybrid varieties under field conditions. The growth of plant shoots is closely associated with the size of the root system. Larger roots are often beneficial for absorbing nutrients, thereby leading to a higher shoot biomass. The size and morphology of the roots affect the absorption of nutrients, and the root length is an important indicator of the effectiveness of root interception to acquire nitrogen. Thus, genotypes that exhibit greater nitrogen uptake have a large root system, especially longer roots (*Li et al., 2017*; *Duan, 2019*). In this study, under the low nitrogen conditions, we observed that the N-efficient maize hybrid ZD958 had longer roots, higher root dry weights and root active absorption area than the N-inefficient maize hybrid QS101 at lower N rates, which was mainly attributable to the longer roots and higher root dry weight in the 20–60 cm soil depth. These findings suggest that the N-efficient maize hybrid had a greater capacity for growing longer roots, which was reflected by the aboveground performance in terms of its greater shoot biomass and grain filling rate compared with QS101. A variety with a greater capacity for growing longer roots can maximize the capture of nitrogen from the soil

(*Garnett, Conn & Kaiser, 2009*; *Mi et al., 2010*), and we found that the nitrogen content and shoot biomass of ZD958 was significantly higher than that of QS101. Studies also have shown that the deep root environment of the soil was relatively stable, which helps to enhance the buffering capacity to the adverse soil environment (*Wasson et al., 2012*), improve the stress resistance, delay root senescence, and maintain root nutrient and water supply to the shoot, obtain high grain yield (*Saengwilai et al., 2014*). Therefore, we conclude that under low nitrogen conditions, larger shoot and root biomasses as well as higher grain filling rates will contribute to better grain yields, and thus higher NUE levels for N-efficient maize hybrids.

The supply of nitrogen affects the growth of roots. Numerous studies have investigated the effects of the nitrogen supply on the growth of maize roots under field conditions (*Tian et al., 2006*; *Wu et al., 2011a*; *Wu et al., 2011b*; *Liu et al., 2017*). Nitrogen application has a positive effect on root growth (*Liu et al., 2009*; *York et al., 2015*) but also adverse effects when provided in excess (*Tian et al., 2008*; *Chen et al., 2015*). Root growth has a parabolic linear relationship with the nitrogen application rate and excessive nitrogen application limits the growth of the root system (*Feng et al., 2016*). Our results are similar to those obtained in previous studies. The length and dry weight of the maize roots increased with the nitrogen application rate, but the length of the roots decreased in the N300 treatment (Figs. 3 and 4), where this difference occurred mainly in the 20–60 cm soil depth. The root system was obviously shallower in the N300 treatment than the N225 treatment and it was mostly distributed in the upper part of the soil. The root length and root dry weight were also small, especially the root length in the N300 treatment. The same trend was observed for both genotypes. This could explain why the maize grain yield did not increase under higher nitrogen conditions.

Previous studies showed that N-efficient maize varieties obtained higher PNUE values (*Echarte, Rothstein & Tollenaar, 2008*; *Chen et al., 2014*). Similar to previous studies, we determined differences in SLN and Pn between the two hybrid varieties. PNUE can be expressed as the ratio of Pn relative to SLN. At the low nitrogen rate, ZD958 had higher SLN and Pn as well as PNUE values, which indicated that the higher PNUE was attributed to a higher Pn value. A higher PNUE can increase the grain filling rate and plant biomass, but also the NUE. The physiological mechanism responsible for the higher PNUE in N-efficient maize hybrids under low nitrogen conditions is not clear. However, it is possible that hybrid varieties with higher PNUE values can allocate less nitrogen to their non-photosynthetic components than hybrids with lower PNUE values (*Hikosaka, 2004*; *Mu et al., 2016*). In this study, we only investigated two varieties that differed significantly in terms of their performance under low nitrogen conditions. The selection of varieties was limited and thus it would be useful to select more varieties for further assessments.

## CONCLUSIONS

In this study, low nitrogen application rates substantially reduced shoot and root growth, nitrogen uptake, and the grain yield in maize. The N-efficient maize hybrid variety obtained a higher grain yield, greater shoot nitrogen uptake to the shoots from the soil, and a higher

NUE at low nitrogen application rates. Higher root and shoot biomasses, a deeper root distribution, longer root length, root active absorption area, and a higher leaf PNUE are beneficial for the grain yield and NUE. Thus, it is possible to obtain a high grain yield and NUE in maize under low nitrogen conditions. The plant traits comprising a higher PNUE and deeper root distribution can be used as selection criteria in breeding programs to select N-efficient hybrid varieties for growth in low nitrogen conditions.

## ACKNOWLEDGEMENTS

We would like to thank the reviewers for helping us to improve our original manuscript, and we are also grateful to Ding Ruixia, Nie Junfeng, and Yang Baoping for field work assistance during experimental period.

### Funding

This study was supported by funding from the National High-Tech Research and Development Programs of China ("863 Program") for the 12th Five-Year Plants (No. 2013AA102902), the Agro-scientific Research in the Public Interest under Grant (201303104), and the National Natural Science Foundation of China (No. 31601256). The funders had no role in study design, data collection and analysis, decision to publish, or preparation of the manuscript.

### Grant Disclosures

The following grant information was disclosed by the authors:
National High-Tech Research and Development Programs of China ("863 Program"): 2013AA102902.
National Natural Science Foundation of China: 31601256.

### Competing Interests

The authors declare there are no competing interests.

### Author Contributions

- Wennan Su conceived and designed the experiments, performed the experiments, analyzed the data, contributed reagents/materials/analysis tools, prepared figures and/or tables, approved the final draft.
- Muhammad Kamran performed the experiments, analyzed the data, contributed reagents/materials/analysis tools, authored or reviewed drafts of the paper.
- Jun Xie conceived and designed the experiments, performed the experiments, contributed reagents/materials/analysis tools, prepared figures and/or tables.
- Xiangping Meng contributed reagents/materials/analysis tools.
- Qingfang Han and Juan Han conceived and designed the experiments, authored or reviewed drafts of the paper, approved the final draft.
- Tiening Liu conceived and designed the experiments, approved the final draft.

## Data Availability

The raw measurements are available as a Supplemental File.

## Supplemental Information

Supplemental information for this article can be found online at http://dx.doi.org/10.7717/peerj.7294#supplemental-information.

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
