# Peer review of "Shoot and root traits of summer maize hybrid varieties with higher grain yields and higher nitrogen use efficiency at low nitrogen application rates"

_PeerJ, doi:10.7717/peerj.7294_

## Round 0.1 · original submission · Major Revisions

Dear authors,

Your paper has been assessed by two reviewers and myself as academic Editor.

As you could see below, the manuscript needs a major revision.
Please address all concerns of reviewer 1 and submit a revised version of the manuscript. Please include a detailed response to each reviewer.
English language should be corrected by a native speaker with scientific expertise.

Please include raw data in excel and also elaborate better figures (more graphical output)

[]

Reviewer 1 ·

Basic reporting

1. The is need for prof reading from the an English speaking person or prof-reading services. There are lots of grammatical errors that need to be addressed before the paper can be considered for publication.
2. On the introduction the author need focus more on the physiology on what has been done in terms of nitrogen use efficiency. see the following papers

Bänziger, M., Setimela, P. S., Hodson, D., & Vivek, B. (2006). Breeding for improved abiotic stress tolerance in maize adapted to southern Africa. Agricultural Water Management, 80(1–3), 212-224

Bänziger, M., G. O. Edmeades, and H. R. Lafitte. "Selection for drought tolerance increases maize yields across a range of nitrogen levels." Crop science 39.4 (1999): 1035-1040.
Bänziger, M., F. J. Betrán, and H. R. Lafitte. "Efficiency of high-nitrogen selection environments for improving maize for low-nitrogen target environments." Crop Science 37.4 (1997): 1103-1109.
Bänziger, M., and H. R. Lafitte. "Efficiency of secondary traits for improving maize for low-nitrogen target environments." Crop Science 37.4 (1997): 1110-1117.

3. From the attached citation the introduction and the background can be improved.

Abstract:
Line 19 remove the background
Line 21 remove the word methods
Line 25 remove the word Results
Line 31 remove the word Discussion

The state the objective of research or the rationale for doing the research. From the abstract state the design used. Proof reading is need and follow the style of the journal.

Introduction:

Line 36 There is no citation for Tester and Kandgride
Line 38-40 State how nitrogen fertilizer has caused problems in agriculture. Line 43 can be combined to show the effects of nitrogen fertilizer

Line 48 there is need for a citation on China being the largest consumer of nitrogen fertilizer

Line 48-60; The paragraph is too long break it into two
Line 81-86 ; Some of the materials should be moved to materials and methods or simply delete

Experimental design

1.mention the season when the experiment was conducted
2. How were the two hybrids identified as NUE and non-NUE.
3. Line 100 can be combine with line 90
4. Line 105 need revision
5. Pn and PNUE- define the abbreviation before use
6. Line 117 its not clear why the midrib is being avoided.
7. Line 135; replace the worked killed
8. On the materials and methods include the linear model for the experiment.
9. What the maturity of the two hybrids

Validity of the findings

1. There is need sure that the experiment is bring in new information.
2. The experiment could have been better if more genotypes were used. That will increase the validity of the experiment and increase the population of inference.

·

Basic reporting

The title of the manuscript and the text provide evidence that are important quality assessors of corn hybrids for breeders. I admit that I am not a professional in English but I understand and technically what I read is fine.
The introduction and background are clear and consistent with the cited quotations. Follow order in the development of the text. I consider it to be within the Peer journal standard.

Experimental design

The materials and methods are widely described in detail with sufficient detail and information to replicate. The statistical procedures are adequate.

Validity of the findings

The result is not novel, however it provides knowledge, the evaluation in hybrid germplasm updates the evaluation methodology. It is possible to evaluate this methodology using others and more hybrids as well as other places. The data are sufficient and strengthen a clear analysis with quotes referred to in the text. Clear conclusions and are supported by data, congruent with the purpose of the manuscript.
I believe that it is important to update references, and to be selective those that really contribute or contrast what has been discussed. References of the last ten years is desirable.

Additional comments

I believe that the manuscript must be accepted and continue with the editorial process until its publication.

---

## Round 0.2 · accepted · Accept

Dear author

I can read that you have addressed all the reviewers concerns. The reviewers comments have been responded adequately.

I congratulate you for the nice piece of work, which will add value to PeerJ.